# The Impact of Coal's Petrographic Composition on Its Suitability for the Gasification Process: The Example of Polish Deposits

**Barbara Bielowicz *** and **Jacek Misiak**

Faculty of Geology, Geophysics and Environment Protection, AGH University of Science and Technology, Al. Mickiewicza 30, 30-059 Kraków, Poland; misiak@agh.edu.pl
* Correspondence: bbiel@agh.edu.pl

**Abstract:** In this paper, we discuss the impact of the rank of coal, petrographic composition, and physico-chemical coal properties on the release and composition of syngas during coal gasification in a $CO_2$ atmosphere. This study used humic coals (parabituminous to anthracite) and lithotypes (bright coal and dull coal). Gasification was performed at temperatures between 600 and 1100 °C. It was found that the gas release depends on the temperature and rank of coal, and the reactivity increases with the increasing rank of coal. It was shown that the coal lithotype does not affect the gas composition or the process. Until 900 °C, the most intense processes were observed for higher rank coals. Above 1000 °C, the most reactive coals had a vitrinite reflectance of 0.5–0.6%. It was confirmed that the gasification of low-rank coal should be performed at temperatures above 1000 °C, and the reactivity of coal depends on the petrographic composition and physico-chemical features. It was shown that inertinite has a negative impact on the $H_2$ content; at 950 °C, the increase in $H_2$ depends on the rank of coal and vitrinite content. The physicochemical properties of coal rely on the content of maceral groups and the rank of coal. An improved understanding these relationships will allow the optimal selection of coal for gasification.

**Keywords:** coal gasification; reactivity; rank of coal; petrography; maceral

## 1. Introduction

In the face of the increasingly restrictive climate and energy policies of the European Union, Poland must look for solutions enabling further use of coal as an energy source. The Polish energy industry has been using coal technologies for decades, and there are no arguments in favor of moving away from them.

Taking into consideration the transition from coal mining and the combustion of coal for energy production, clean-coal technologies are being actively developed. These include the gasification process, which, depending on the technology used, can act as an effective way to reduce carbon dioxide emissions. The reduction in $CO_2$ produced is a key issue when obtaining energy from solid fuels. Therefore, research into the possibilities of using coal in technologies other than direct combustion is of key importance.

Globally, coal units generate about 41% of electricity. In China it is 78%, in India it is 68%, and in the USA it is 46%. Within the Energiewende, Germany generates over 40% of its energy in coal power plants [1]. According to the International Energy Agency, by 2030 the capacity installed in coal power plants around the world will increase by approximately 50% [2]. In some countries, especially in Asia, it will increase many times over. In spite of official data pointing in exactly the opposite direction, the European Union currently produces 137 GW of energy from coal, and in a dozen or so years this

production will increase to 177 GW. The increase in installed capacity in coal units worldwide is worth noting: India from 92 GW to 267 GW, the Association of Southeast Asian Nations (ASEAN) countries from 52 GW to 142 GW, South Korea, Australia, and New Zealand (total) from 53 GW to 60 GW, North America from 354 GW to 369 GW, and Russia from 43 GW to 47 GW. There is a large market for clean coal technologies, which will reduce the environmental impact [2].

In Poland, approximately 70% of energy is obtained from solid fuels (bituminous coal and lignite). Despite the ongoing efforts to build a nuclear power plant and increase the share of renewable energy sources, a significant reduction in coal-based energy is not likely to happen in the next few years. Therefore, more and more attention has been focused on clean coal technologies. These include the gasification of bituminous coal [3].

The development of gasification technology is particularly dynamic in countries with high economic potential that do not have their own reserves of oil and natural gas. These are countries of the Asia and Oceania region, including China, which have now taken a leading position in the development of coal technologies. Fixed bed reactor technology has the largest share in gas production by coal gasification (57% of syngas produced), mainly due to the production capacity of SASOL plants in South Africa; however, this technology is no longer being developed [2]. Taking into account both operating and currently developed gasification plants, it can be concluded that the continuously developed entrained flow (pulverized) coal gasification technologies are still dominant when it comes to the gas production from coal gasification. The most popular gasification technologies using entrained flow reactors include Shell, GE/Texaco, Siemens and, thanks to rapid development in recent years, the ECUST Opposed Multi-Burner (OMB) technology (East China University of Science and Technology). One way to reduce investment and operating costs and improve the reliability and operation of the reactor is the development of low temperature gasification technologies in a $CO_2$ atmosphere. The advantage of this solution lies in its use of an additional carbon dioxide stream as a gasification agent that feeds coal (C element) and oxygen and improves the efficiency of the process. The use of $CO_2$ as a gasifying agent is possible due to the Boudouard reaction, whose product is carbon monoxide, which is, alongside hydrogen, the basic component of synthesis gas [4]. Research on the possibility of using $CO_2$ as a gasifying agent has been conducted since the second half of the 20th century [5]. The research has covered various aspects of this process: the impact of individual coal parameters, the efficiency of the process, and the impact on the natural environment. The positive impact on the environment and the reduction in greenhouse gas emissions, such as carbon dioxide, are the reasons gasification using $CO_2$ as a gasifying agent is currently being developed. So far, there have been no large installations for gasification of coal under a $CO_2$ atmosphere, although this technology is being developed in numerous countries around the world [4,6–9]. According to previous research, the introduction of carbon dioxide into the coal gasification reactor results in an increase in the degree of reaction of carbon and the amount of carbon monoxide in the synthesis gas, when compared to classical gasification under an air atmosphere. Carbon dioxide supplied to the system is also used as an oxygen carrier, which allows for a significant reduction in its consumption [4].

The clean coal technology market in countries such as Poland, where 90% of electricity is generated by coal units, could be very attractive for companies with clean technologies [10]. Clean coal technology projects could likewise be attractive to countries such as Poland, which have their own energy sources and can be energy independent. One example is a joint venture of Enea and Mitsubishi Hitachi Power Systems, which aims to build a coal-fired power plant at the Bogdanka mine. The 500 MW power plant will be one of the largest of this type in the world and will use gas from the gasification of coal.

Such an undertaking is an opportunity to meet the emission limit of 550 g $CO_2$ of fossil fuel origin per kWh of electricity, set by the European Union [11]. Therefore, it was decided to investigate the possibility of using Polish coal in a process that reduces $CO_2$ emissions.

However, in order to examine such technologically advanced processes, a deep knowledge of the fundamental relationships between the petrographic composition, physico-chemical properties of coal, and its reactivity during the gasification process is needed. This paper focuses on the influence

of lithological development on coal's ability to convert during gasification, which is understood as the reactivity of the coal. The questions of reactivity and petrographic composition have been examined by many researchers [12,13]; however, generally the results apply only to the selected deposit, without taking into account the individual lithotypes of coal. Reactivity can be used as a simple and reliable method to compare coal types. What is more, the effect of the catalyst or mineral matter on pyrolysis and gasification processes can also be determined based on changes in the reactivity of coal. The comparison of curves of the carbon conversion rate [14] is among the most common methods allowing determination of the reactivity of coal. The half-conversion time $\tau 0.5$, understood as the time after which 50% of the elemental C will be converted, is determined using the conversion rate of elemental carbon [15]. This study analyzed the composition and amount of produced syngas; conversion rates were not calculated. The impact of petrographic composition, particle size, coal rank, and the ash content on the reactivity of coal is widely accepted [16]. In most cases, the thermochemical processing depends on the type of mineral matter. The catalytic effect of alkalis has been confirmed by numerous studies. The arrangement of the components in the organic matter is also of great importance. A high dispersion of the components of ash is the reason why even small amounts of ash may have a stronger impact on the reactivity of coal than greater amounts at one location [17]. The mineral matter can block the pores; on the other hand, it has a positive impact on their development during gasification. In addition, mineral matter has an impact on the chemical structure of coal. The structure of organic matter (and therefore carbon active centers) is different depending on the occurrence of specific elements, such as calcium [18]. Therefore, petrographic studies are required to determine the type and distribution of mineral matter. According to other researchers, reactivity heavily depends on the petrographic composition of coal [19].

The conducted research is unique as it covers a wide range of coals with different ranks of coal and petrographic compositions, ranging from subbituminous coal to anthracite, which were tested using advanced methods. This paper discusses the impact of the petrographic and chemical composition, and technological parameters of coal, on the quantity and composition of generated syngas. An experiment examining the possibility of gasification of coal of different ranks in a $CO_2$ atmosphere was performed at the Central Measurement and Testing Laboratory in Jastrzębie-Zdrój. The obtained results will allow the selection of potential deposits that are best suited for gasification in a $CO_2$ atmosphere.

## 2. Experiment

### 2.1. Samples

This study used samples from Polish deposits. The location of deposits is presented on a map (Figure 1). The examined samples, with the exception of anthracite from Vietnam (the Vietmindo deposit in the Quang Yen Basin), were collected from Polish bituminous coal seams of the Upper Silesian Coal Basin and Lublin Coal Basin. Both basins are of Carboniferous age and were formed in the Variscan orogeny. Coal deposits occur in molasse formations formed as sedimentary deposits: conglomerates, sandstones, mudstones, and clays (Figure 1). The coal is characterized by a highly variable petrographic composition and degree of coalification.

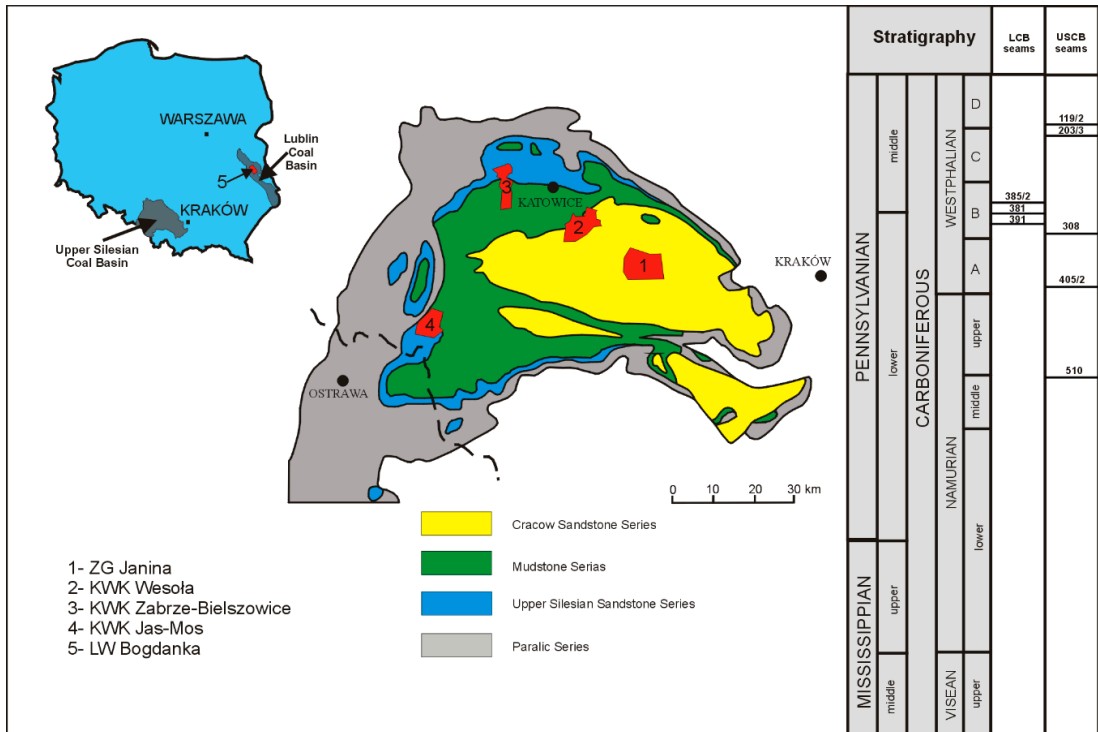

**Figure 1.** Location of coal deposits used for sampling (map of the USCB-based [20] and lithostratigraphic division of the Upper Silesian Basin (USCB) and Lublin Coal Basin (LCB) [20,21]).

The study used 18 samples of humic coal of different degrees of coalification, ranging from bituminous coal to anthracite. Bituminous coal was macroscopically described according to Diessel [22]. The following lithotypes were determined: (B) Bright coal, less than 10% dull laminae; (BB) Banded bright coal, 10–40% dull laminae; (BC) Banded coal, dull and bright laminae in equal proportions; (BD) Banded dull coal, 10–40% bright laminae; (D) Dull coal, less than 10% bright laminae; (F) Fibrous coal. The petrographic analysis used polished sections prepared in accordance with the ISO 7404-2:2009 standard. The petrographic examination and reflectance measurements were carried out under reflected white and blue light, with the use of a Zeiss Opton microscope and in accordance with the ISO 7404-standard. The nomenclature and determination of macerals from the vitrinite group were based on the guidelines of the International Committee for Coal and Organic Petrology [23,24]. The macerals of the liptinite group were determined, classified, and described according to the guidelines by Pickel et al. [25].

All proximate and ultimate analyses were performed in the Central Measurement and Testing Laboratory in Jastrzębie-Zdrój, according to ISO standards. The moisture, ash content, volatile matter, net calorific value, and the elemental content of C, H, N, and O were determined in the examined coal.

### 2.2. The Gasification of Coal to Carbon Dioxide

The coal gasification took place in the temperature ranges of 600–1100 °C, and the gasifying agent was carbon dioxide. The experiment was carried out using a specially designed apparatus [26,27], and coal with a grain size between 0.2 mm and 3.0 mm. The reason for the lower limit was the sieve bottom of the reactor (with a square mesh of 0.16 mm). The upper grain limit corresponds to a typical laboratory sample, prepared according to Polish standards for physical and chemical analyses. Bituminous coal is characterized by highly variable total water content. In the as-received bituminous coal, the water content ranges between 2% and 15%. To reduce the influence of the water content, the tested samples were dried to an air-dry state. Prior to grinding, the samples were thoroughly mixed. To minimize losses in dust with a grain size below 0.2 mm, the grinding process was performed

in several stages. First, the <10 mm fraction was removed, then the sample was passed through a jaw crusher several times; each time, the <10 mm fraction was sieved. The collected fractions were combined and ground to a size below 3 mm using the same method. After grinding, they were evenly spread, dried at 40 °C, and stored in a laboratory atmosphere for four hours, until they reached a constant weight (equilibrium with atmospheric moisture). A subsample for physicochemical analysis and 100 g subsamples for reactivity studies were isolated from the laboratory samples in the air-dry state. To recover the desired fraction (0.2 mm–3.0 mm), the aforementioned subsamples were sieved using a 0.2 mm sieve [26]. The analysis used an averaged 1 g sample of coal. The reactivity was determined in the reactor presented in Figure 2. The process was described in detail in a separate publication [26,27]. In the first stage, the reactor was heated up to 600 °C for drying and initial degassing. After reaching the temperature of 600 °C, $CO_2$ was supplied to the system as a gasifying agent. The $CO_2$ inlet gas flow was set at 15 dm$^3$/h. The maximum heating time to 1100 °C was 19 min. During this time, two gas samples from the temperature ranges of 600–950 °C and 950–1100 °C were collected. Gas composition was measured using a gas chromatograph.

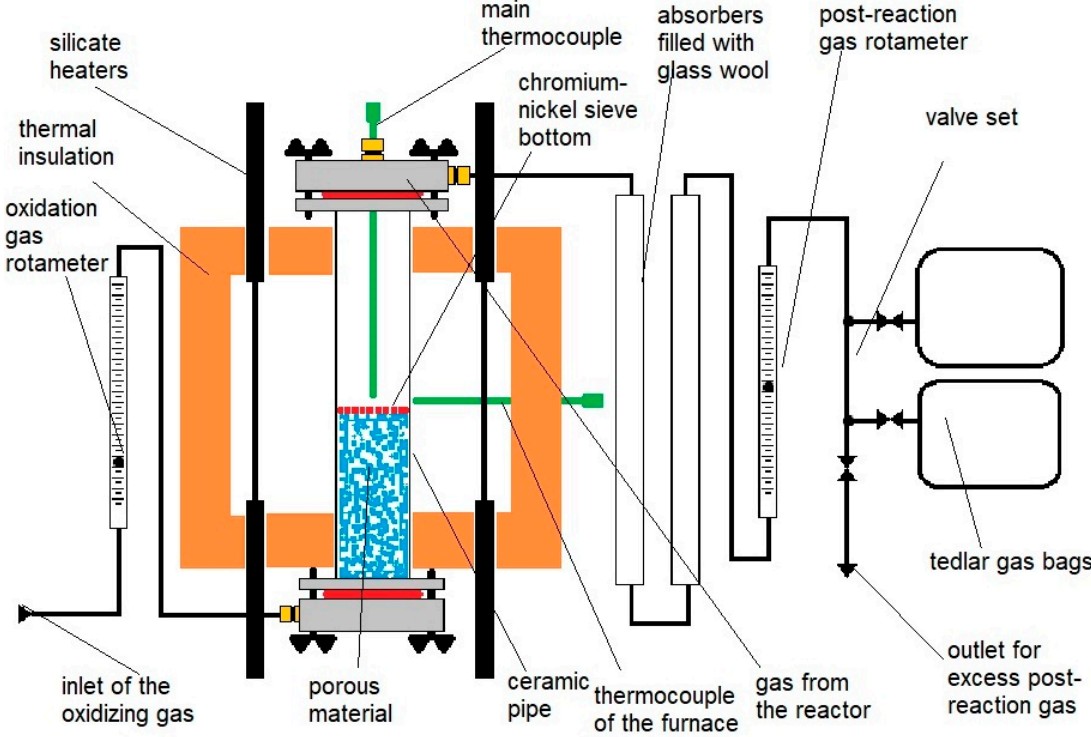

**Figure 2.** The reactor for reactivity analysis [26].

A total of 36 syngas samples were taken. The percentages of $H_2$, $O_2$, CO, $CO_2$, $CH_4$, $N_2$, and $H_2S$, in addition to the gas density at 25 °C, were determined for each sample. The gas density during the gasification was constantly monitored.

In order to investigate the relationship between the parameters of coal and the amount and composition of the obtained syngas, basic and descriptive statistical analysis was carried out by determining the average, minimum, maximum, standard deviation, and coefficient of variation and skewness. The results were correlated, and statistical analysis was performed using Statistica10 software. The purpose of the correlation analysis was to determine the relationships between the variables: their strength, form, and directions. Assuming a normal distribution of results, the Pearson test was used to assess the degree of correlation. The significance of the correlation was verified by means of a statistical test and rejection of the null hypothesis. A significance level of α = 0.05 was adopted.

### 3. The Characteristics of the Examined Coal

The coal collected from the sampling sites was highly variable both in terms of the degree of coalification (reflectance in the range of 0.5–1.1%) and lithological development (the samples dominated by bright coal (vitrain coal) were compared with samples with a dominant share of dull coal (durain coal)). The petrographic composition of coal was dependent on its lithological development (Table 1). Coals dominated by bright coal (vitrain coal) had a dominant share, ranging from 69.2 to 95%, of microscopic components from the vitrinite group, among which the most common was collotelinite. Macerals of liptinite and inertinite groups were present in marginal amounts. The macerals of the liptinite and inertinite groups ranged from 4.8 to 29.2%, and 5.2 to 13.4%, respectively. In the case of coals with a dominant share of dull coal (durain) determined macroscopically, a much lower proportion of macerals of the vitrinite group, ranging from 40 to 55.4%, has been observed. The predominant component in the vitrinite group was collodetrinite. The other two maceral groups have shown great diversity. The macerals of the liptinite group ranged from 19.4 to 38.2%, while macerals from the inertinite group accounted for 11.2 to 28.2%. The mineral matter content was low, and ranged from less than 1% to 7.6%.

Moisture is one of the main parameters of coal quality. The total moisture content in the tested samples decreased with the increasing rank of coal (15.8% in subbituminous/bituminous coals to 0.8% in coking coal) (Table 2). The volatile matter decreased with the rank of coal from 40.5% in the No. 119 seam to 4.2% in anthracite. The ash content (dry basis) ranged from 1.2% to 24.14%. In selected seams, the ash content was noticeably higher in the case of dull coal. The sulfur content (dry basis) was not clearly related to the lithotype or coal rank and ranged between 0.3% and 5.3%. The carbon content (dry, ash-free basis) varied with the coal rank between 74.07% and 93.87%.

The examined coals were para-, ortho- and meta-bituminous coal and para-anthracite. The suitability of Polish lignite for gasification was discussed in Bielowicz [26]. This study was aimed at determining the amount and composition of gases released from coal in the heating process. The first phase of the experiment, up to 400 °C, involved mainly evaporation, while thermal degradation was less pronounced; furthermore, water, volatile organics, hydrocarbons, and aromatic hydrocarbons, including benzene, toluene, and HCl, were released. A low-temperature pyrolysis process was carried out at a temperature of 400–600 °C (the samples were heated without an oxidizing agent ($CO_2$)). Gasification with $CO_2$ as the gasifying agent started at 600 °C. The main phase of thermal decomposition (the low-temperature carbonization area), observed during heating to a temperature of about 750 °C, produced mainly gas and char (accompanied by smaller amounts of tar, liquor, and hydrogen) [28]. According to the author of [28], medium temperature carbonization can be observed up to 900 °C. The resulting products include gas, char, tar, liquor, and hydrogen. High temperature carbonization (1100 °C) involves significant thermal cracking of volatiles. As shown in many works [29–31], this is crucial in coal conversions.

Due to the fact that the temperatures at which the experiment was conducted reached up to 1100 °C, it was assumed that pyrolysis takes place at a temperature below 900 °C, while the gasification process begins above this temperature [32–34]. It has been suggested that the pyrolysis and gasification processes in a $CO_2$ atmosphere take place simultaneously, and the C–$CO_2$ reaction is difficult to detect at temperatures below 800–900 °C [35]. Therefore, it was assumed that the first stage is pyrolysis; char–$CO_2$ reactions take place only after its completion.

**Table 1.** Petrographic composition of the examined coal.

| Sample No | Deposit | Seam No. | Lithotype | Telinite | Collotelinite | Collodetrinite | Vitrodetrinite | Corpogelinite | Gelinite | Macrosporinite | Microsporinite | Cutinite | Resinite | Liptodetrinite |
|---|---|---|---|---|---|---|---|---|---|---|---|---|---|---|
| 1 | Jas-Mos | 510 | Coking coal | 0 | 6.5 | 70 | 0 | 0 | 0 | 0 | 0 | 0 | 0 | 0 |
| 2 | Vietnam | Anthracite | 0 | 100 | 0 | 0 | 0 | 0 | 0 | 0 | 0 | 0 | 0 | |
| 3 | Janina | 119/2 | Bright coal | 1.2 | 40.4 | 34.6 | 0.6 | 0.8 | 0.2 | 0.6 | 5.2 | 0.4 | 0.2 | 1.4 |
| 4 | Janina | 119/2 | Dull coal | 0.6 | 11.4 | 23 | 3.4 | 0.8 | 1.2 | 2.4 | 16.4 | 1 | 1 | 7.4 |
| 5 | Janina | 203/2 | Bright coal | 0.4 | 22.6 | 41 | 1 | 0.4 | 3.4 | 0.8 | 3 | 0.4 | 0.2 | 0.8 |
| 6 | Janina | 203/2 | Dull coal | 0.4 | 12 | 36.2 | 2 | 3.2 | 1.6 | 0.6 | 7.8 | 2.2 | 0.4 | 2.8 |
| 7 | Bielszowice | 510 | Bright coal | 0.2 | 2.8 | 58.2 | 0.2 | 0.2 | 2.6 | 0.6 | 4.4 | 0.6 | 0.6 | 0 |
| 8 | Bielszowice | 510 | Dull coal | 0.6 | 2.6 | 44.6 | 0.4 | 0.4 | 1.8 | 0.6 | 8.8 | 0.2 | 1.2 | 0.4 |
| 9 | Bogdanka | 391 | Bright coal | 0 | 6.6 | 76.8 | 0.4 | 0.8 | 0.4 | 0.2 | 4.6 | 1 | 1 | 0.8 |
| 10 | Bogdanka | 391 | Dull coal | 0.4 | 4.2 | 40.8 | 1.4 | 0.6 | 0.2 | 2.2 | 13 | 1.2 | 2.2 | 8.8 |
| 11 | Bogdanka | 381 | Bright coal | 0.2 | 8.2 | 70.6 | 0.4 | 0.8 | 0.6 | 0.4 | 6.2 | 1.2 | 0.4 | 1.4 |
| 12 | Bogdanka | 381 | Dull coal | 0.2 | 3.2 | 46.6 | 0.2 | 2.8 | 1 | 0.6 | 13 | 4.6 | 1 | 6.8 |
| 13 | Bogdanka | 385/2 | Bright coal | 0.2 | 16.8 | 54.6 | 0.6 | 2 | 1 | 2.6 | 8.6 | 0.8 | 0.2 | 1.2 |
| 14 | Bogdanka | 385/2 | Dull coal | 2.2 | 20.8 | 20.2 | 0.8 | 0.6 | 2.2 | 7.8 | 12.8 | 1.2 | 1.4 | 3.8 |
| 15 | Wesoła | 308 | Bright coal | 0.8 | 14.6 | 58.6 | 1 | 3.6 | 0.2 | 1.6 | 5.6 | 1.4 | 0.6 | 0.4 |
| 16 | Wesoła | 308 | Dull coal | 0.4 | 2.2 | 31.4 | 1.8 | 2.6 | 1.6 | 3 | 12.4 | 0.8 | 0.2 | 10.4 |
| 17 | Bielszowice | 405/2 | Bright coal | 1.8 | 4.8 | 53.6 | 5.8 | 1.2 | 2 | 0.2 | 6.4 | 0.2 | 0.8 | 0.6 |
| 18 | Bielszowice | 405/2 | Dull coal | 1 | 4.6 | 30.2 | 2.8 | 1.2 | 2.2 | 1 | 12.8 | 0.6 | 1.6 | 2.6 |

**Table 1.** *Cont.*

| Sample No | Fusinite | Semifusinite | Funginite | Macrinite | Micrinite | Secretinite | Inertodetrinite | Sulfides | Carbonates | Clay Minerals | Vitrinite | Liptinite | Inertinite | Mineral Matter | Reflectance % |
|---|---|---|---|---|---|---|---|---|---|---|---|---|---|---|---|
| 1 | 18.5 | 0 | 1.5 | 0 | 0 | 0 | 3 | 0.5 | 0 | 0 | 76.5 | 23 | 0 | 0.5 | 1.1 |
| 2 | 0 | 0 | 0 | 0 | 0 | 0 | 0 | 0 | 0 | 0 | 100 | 0 | 0 | 0 | 2.3 |
| 3 | 3.6 | 2.4 | 2 | 0.2 | 0.4 | 0.2 | 4.4 | 0.6 | 0 | 0.6 | 77.8 | 13.2 | 7.8 | 1.2 | 0.5 |
| 4 | 4.2 | 6.6 | 4.6 | 3.8 | 0.4 | 0 | 10.8 | 0.8 | 0 | 0.2 | 40.4 | 30.4 | 28.2 | 1 | 0.5 |
| 5 | 5.6 | 8.6 | 1.4 | 1 | 0.4 | 0 | 7.4 | 0.6 | 0 | 1 | 68.8 | 24.4 | 5.2 | 1.6 | 0.5 |
| 6 | 3 | 3.8 | 7.8 | 0.2 | 0.2 | 0.2 | 8 | 1.4 | 0 | 6.2 | 55.4 | 23.2 | 13.8 | 7.6 | 0.5 |
| 7 | 2 | 2.4 | 10.4 | 3.8 | 0.6 | 0.2 | 9.8 | 0 | 0 | 0.4 | 64.2 | 29.2 | 6.2 | 0.4 | 0.7 |
| 8 | 3.2 | 2.4 | 8.4 | 4 | 0.2 | 0.2 | 17.6 | 0.6 | 0 | 1.8 | 50.4 | 36 | 11.2 | 2.4 | 0.7 |
| 9 | 0.2 | 0.8 | 1 | 0.2 | 0.8 | 0 | 1.8 | 2 | 0 | 0.6 | 85 | 4.8 | 7.6 | 2.6 | 0.6 |
| 10 | 5.6 | 1.6 | 2.4 | 1.2 | 0.4 | 0 | 9.6 | 3.8 | 0 | 0.4 | 47.6 | 20.8 | 27.4 | 4.2 | 0.6 |
| 11 | 0.6 | 2 | 3 | 0.2 | 0.2 | 0.2 | 2.8 | 0 | 0 | 0.6 | 80.8 | 9 | 9.6 | 0.6 | 0.6 |
| 12 | 1 | 2.8 | 4.6 | 0.8 | 0.8 | 0 | 9.4 | 0.4 | 0 | 0.2 | 54 | 19.4 | 26 | 0.6 | 0.6 |
| 13 | 2.4 | 2.2 | 2.8 | 0.4 | 0 | 0.2 | 2.2 | 0.8 | 0 | 0.4 | 75.2 | 10.2 | 13.4 | 1.2 | 0.6 |
| 14 | 4 | 7.2 | 2.4 | 1 | 1 | 0.2 | 8.8 | 0.2 | 0 | 1.4 | 46.8 | 24.6 | 27 | 1.6 | 0.6 |
| 15 | 1.8 | 2.4 | 2.4 | 0.2 | 0 | 0 | 1.6 | 0 | 0.4 | 2.8 | 78.8 | 8.4 | 9.6 | 3.2 | 0.6 |
| 16 | 6.2 | 3 | 6.8 | 1.8 | 0.2 | 0.2 | 13 | 0.4 | 0 | 1.6 | 40 | 31.2 | 26.8 | 2 | 0.6 |
| 17 | 7 | 3.2 | 5 | 0.6 | 0 | 0 | 6.8 | 0 | 0 | 0 | 69.2 | 22.6 | 8.2 | 0 | 0.7 |
| 18 | 5 | 5.6 | 8.8 | 3.4 | 0.4 | 0 | 15 | 0 | 0 | 1.2 | 42 | 38.2 | 18.6 | 1.2 | 0.7 |

**Table 2.** Proximate and ultimate analysis of the examined coal.

| Sample No. | Moisture, $M_t^{ar}$ [%] | Ash Content, $A^{db}$ [%] | Volatile Matter, $V^{daf}$ [%]$^f$ | Sulphur Content, $S_t^{db}$ [%] | Net Calorific Value, NCV [MJ/kg] | Carbon Content, $C^{daf}$ [%] | Hydrogen Content, $H^{daf}$ [%] | Nitrogen Content, $N^{daf}$ [%] | Oxygen Content, $O^{daf}$ [%] |
|---|---|---|---|---|---|---|---|---|---|
| 1 | 0.8 | 6.5 | 21.95 | 1.13 | 32.662 | 89.73 | 4.76 | 1.11 | 3.21 |
| 2 | 1.3 | 1.2 | 4.16 | 0.3 | 33.936 | 93.87 | 3.17 | 1.01 | 1.64 |
| 3 | 15.1 | 8.6 | 39.25 | 2.58 | 22.906 | 76.65 | 4.82 | 1.23 | 14.48 |
| 4 | 12.6 | 6.23 | 40.54 | 1.6 | 24.810 | 77.92 | 5.11 | 1.26 | 14.00 |
| 5 | 15.8 | 7.75 | 35.36 | 1.38 | 22.970 | 77.59 | 4.71 | 1.15 | 15.05 |
| 6 | 11.7 | 24.14 | 38.83 | 1.04 | 19.383 | 76.02 | 5.10 | 1.15 | 16.37 |
| 7 | 1.5 | 3.88 | 27.79 | 0.53 | 32.831 | 87.66 | 4.91 | 1.55 | 5.33 |
| 8 | 1.1 | 2.92 | 27.96 | 0.64 | 33.193 | 88.22 | 4.93 | 1.58 | 4.61 |
| 9 | 4 | 8.34 | 39.49 | 3.02 | 28.216 | 80.40 | 5.25 | 2.02 | 9.04 |
| 10 | 3.3 | 11.25 | 39.77 | 5.28 | 27.702 | 79.38 | 5.37 | 1.81 | 7.48 |
| 11 | 5.1 | 3.31 | 35.09 | 1.24 | 29.918 | 81.90 | 5.22 | 1.96 | 9.64 |
| 12 | 4.9 | 3.73 | 35.16 | 1.49 | 29.356 | 81.14 | 5.20 | 1.99 | 10.13 |
| 13 | 4.3 | 6.02 | 38.02 | 1.34 | 29.300 | 81.21 | 5.38 | 1.71 | 10.27 |
| 14 | 3.4 | 10.55 | 37.84 | 1.22 | 27.925 | 81.73 | 5.34 | 1.49 | 10.07 |
| 15 | 6.2 | 10.39 | 39.98 | 0.93 | 23.253 | 74.10 | 4.42 | 1.28 | 19.15 |
| 16 | 6.2 | 15.14 | 38.19 | 0.72 | 22.589 | 74.07 | 4.47 | 1.30 | 19.31 |
| 17 | 1.6 | 10.46 | 32.05 | 0.38 | 29.744 | 85.15 | 5.12 | 1.40 | 7.90 |
| 18 | 1.5 | 10.06 | 33.67 | 0.33 | 30.215 | 85.52 | 5.29 | 1.41 | 7.41 |

The influence of temperature increase on the amount of gas produced was examined by the volume of gas changes with increasing temperature in the gasification reactor (Figures 3–5). The largest flow rates for all samples were observed for temperatures between 600 and 900 °C, and then, at around 950 °C, the volume of gas decreased. The second phase, when the gas volume increases, is the period when the temperature in the gas generator is between 950 and 1100 °C. In the first phase, so-called dirty gas is produced [36]. This gas is composed of gas products and tar vapor, resulting from oxygen-free decomposition of solid fuels. The decrease in the amount of produced gas to the lowest values around the gasification temperature of 950 °C is associated with the transition to high temperature carbonization. This decrease was recorded at a gasification temperature of around 750–800 °C, which is associated with a reduction in the amount of $CO_2$ generated [36]. In addition, it was found that the rank of coal affects the degree of flow rate reduction. In the case of lower rank coals, such as those from the Janina seam, the flow rate reduction was up to 22 dm$^3$/h, while for anthracite it was up to 19 dm$^3$/h. The rank of coal also affected the volume of gas generated in the temperature range between 950 and 1100 °C. The amount of gas produced from anthracite or coking coal was lower than the amount produced from parabituminous coal. This is associated with the amount of volatile components in the tested coals. The volatile matter content decreased with the increasing rank of coal (Figure 3). According to the authors of [19,37,38], the amount of volatiles affects the reactivity of coal. The reasons for this include changes in the organic structure of carbonaceous matter as a result of coalification, a reduction in heteroatoms, and enlarged heat-resistant aromatic cores of structural units or smaller shares of the peripheral part. During the second stage, CO, $CO_2$, and small quantities of methane were released. The relationship between pyrolysis and gasification was discussed by Jüntgen and Heek [39]. If the heating process is slow, pyrolysis takes place at temperatures from about 350 °C. However, at these temperatures, the gasification process for volatiles or char with steam takes place slowly. An increase in the concentration of volatiles outside the particle can be observed. In addition, gasification starts once the devolatilization process is over. At a high rate of heating, pyrolysis and gasification take place at the same time and, as a result, it is not possible for high concentrations of volatiles to build up [39]. The tests were carried out at a relatively low temperature increase rate. Therefore, phases with increased gas flow could be clearly observed. Heterogeneous reactions with coal, Boudouard, and hydrogenation reactions were mainly observed during the second stage of the experiment, which was the gasification process [33].

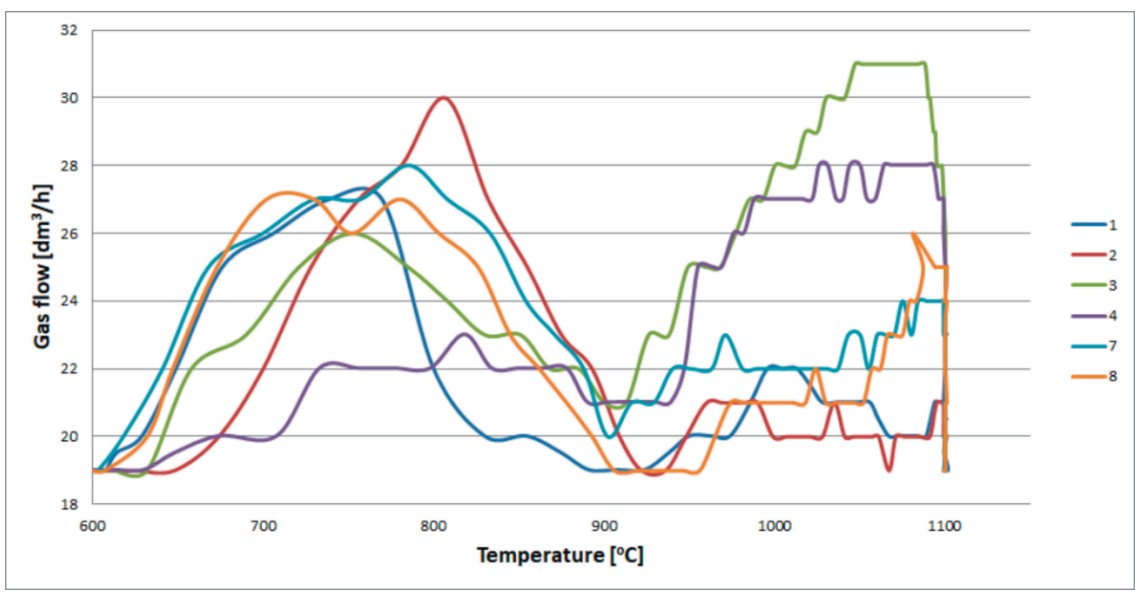

**Figure 3.** Gas flow [dm$^3$/h] as a function of temperature for the USCB coals with a different rank of coal and lithotype. 1: Jas-Mos, seam 510, coking coal; 2: Vietnam, Anthracite; 3: Janina, seam 119/2, bright coal; 4: Janina, seam 119/2, dull coal; 7: Bielszowice, seam 510, bright coal; 8: Bielszowice, seam 510, dull coal.

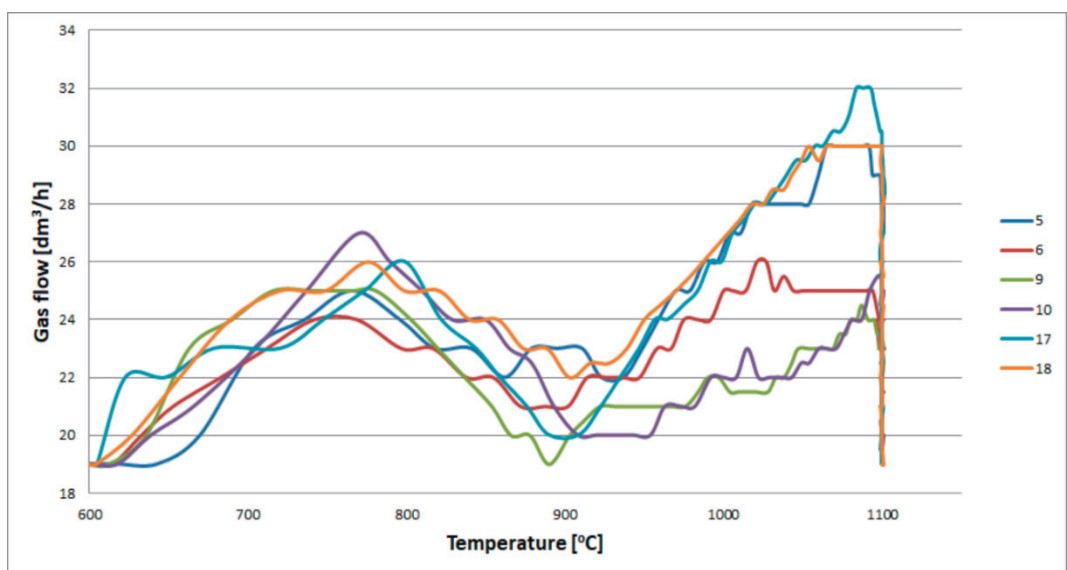

**Figure 4.** Gas flow [dm$^3$/h] as a function of temperature for the USCB coals with a vitrinite reflectance in the range of 0.5–0.7%. 5: Janina, seam 203/2, bright coal; 6: Janina, seam 203/2, dull coal; 9: Bogdanka, seam 391, bright coal; 10: Bogdanka, seam 391, dull coal; 17: Bielszowice, seam 405/2, bright coal; 18: Bielszowice, seam 405/2, dull coal.

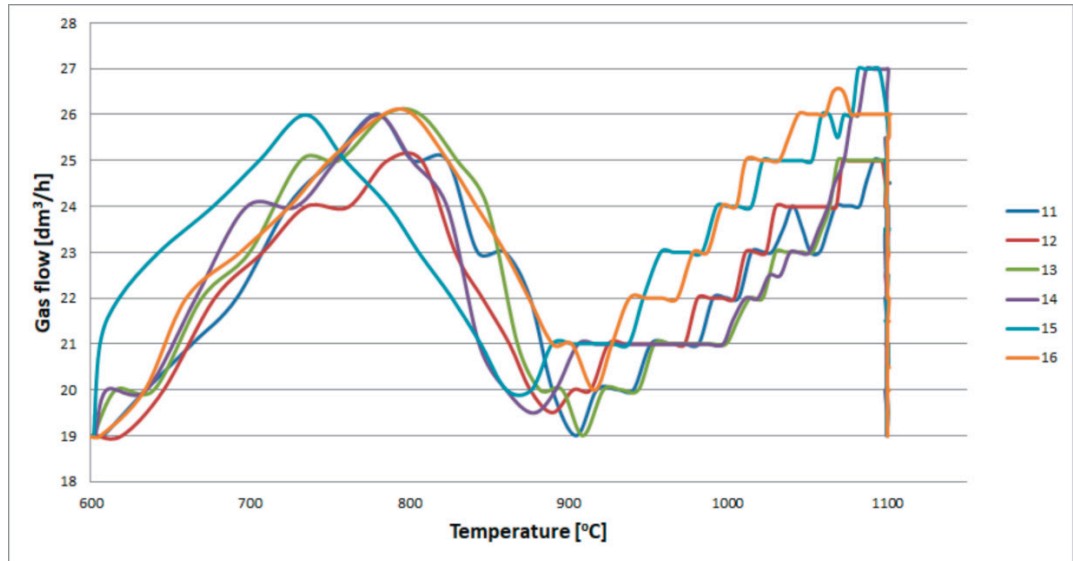

**Figure 5.** Gas flow [dm³/h] as a function of temperature in coal from the LCB (Lublin Coal Basin). 11: Bogdanka, seam 381, bright coal; 12: Bogdanka, seam 381, dull coal; 13: Bogdanka, seam 385/2, bright coal; 14: Bogdanka, seam 385/2, dull coal; 15: Wesoła, seam 308, bright coal, 16: Wesoła, seam 308, dull coal.

When it comes to coals with a higher rank (anthracite, coking coal), larger amounts of gas were released up to a temperature of around 900 °C, while at a temperature of about 1000 °C, the gas flow was clearly lower than for coals with a lower rank of coal. The reverse flow pattern is observed for coal with a reflectance of approximately 0.5%, collected from the Janina coal mine. In this case, the higher gas flow can be observed at higher temperatures (Figures 3 and 4). For coal with a reflectance of 0.7% collected from the Bielszowice deposit, the gas flow at a temperature of about 1000 °C was clearly lower than at 750–800 °C (Figure 4). When comparing coal samples from individual seams (with a similar degree of coalification) within the deposit, slight differences in the gas flow can be observed. The influence on the maximum gas flow has not been confirmed (Figure 5). Therefore, it can be concluded that the impact of the petrographic composition on the amount of released gas is limited; meanwhile, the gas flow is affected by other factors.

The proportions of $H_2$, $O_2$, CO, $CO_2$, $CH_4$, $N_2$, and $H_2S$ were measured in the samples collected at temperatures up to 950 °C and 1050 °C (Table 3).

At a temperature up to 950 °C, the hydrogen generation in the examined coals was constant at approximately 2%, with the exception of samples of a higher rank, namely coking coal and anthracite, for which these values were much higher at 9% and 15%, respectively (Figure 6). At a temperature up to 1050 °C, the amount of generated hydrogen dropped to a value in the range of 1–1.5% for all of the examined samples. The amount of CO generated at a temperature up to 950 °C ranged from 12% in anthracite to 52% in coals with a lower rank of coal. This was lower than the amount of CO generated at 1050 °C, which was also the lowest for anthracite, while when it came to coals of a lower rank, it reached up to 76%. For low rank coals, the highest amounts of nitrogen were formed at a temperature of about 950 °C; when it came to coals with a higher rank of coal, higher amounts of nitrogen were generated at a temperature of about 1050 °C. In both cases, these amounts were highly variable, ranging from a few to several percent. The amount of oxygen generated in both temperature ranges was small, reaching up to 3%. Carbon dioxide was generated in large quantities in both temperature ranges: up to 42–75% at a temperature of up to 950 °C, and 17–70% at a temperature of 1050 °C. The maximum total quantity was produced in anthracite. The amount of methane generated at a temperature of up to 950 °C was much higher than for a temperature of 1050 °C; it should be noted that this amount was below 1% in all cases.

**Table 3.** The chemical composition of syngas.

| Sample No | Temperature | $H_2$ | $O_2$ | CO | $CO_2$ | $CO + CO_2$ | $CH_4$ | $N_2$ | $H_2S$ | Gas Density at 25 °C |
|---|---|---|---|---|---|---|---|---|---|---|
| | °C | % | % | % | % | % | % | % | % | kg/m³ |
| 1 | 950 | 8.95 | 0.17 | 20.52 | 67.66 | 88.18 | 0.35 | 2.35 | 0.05 | 1.50 |
| | 1050 | 1.38 | 0.10 | 39.30 | 58.50 | 97.8 | 0.10 | 0.62 | 0.01 | 1.52 |
| 2 | 950 | 15.18 | 0.45 | 11.34 | 72.31 | 83.65 | 0.50 | 0.22 | 0.04 | 1.46 |
| | 1050 | 1.55 | 0.21 | 28.15 | 69.88 | 98.03 | 0.01 | 0.20 | 0.01 | 1.59 |
| 3 | 950 | 2.16 | 0.14 | 45.68 | 48.24 | 93.92 | 0.25 | 3.53 | 0.02 | 1.44 |
| | 1050 | 1.47 | 0.24 | 75.86 | 16.37 | 92.23 | 0.10 | 5.96 | 0.01 | 1.24 |
| 4 | 950 | 2.61 | 0.47 | 52.20 | 42.08 | 94.28 | 0.16 | 2.48 | 0.01 | 1.40 |
| | 1050 | 1.26 | 0.31 | 73.48 | 20.42 | 93.9 | 0.11 | 4.42 | 0.01 | 1.27 |
| 5 | 950 | 2.28 | 0.17 | 48.37 | 46.02 | 94.39 | 0.24 | 2.92 | 0.03 | 1.43 |
| | 1050 | 1.63 | 0.26 | 75.29 | 19.09 | 94.38 | 0.09 | 3.64 | 0.01 | 1.25 |
| 6 | 950 | 2.67 | 0.36 | 42.21 | 53.37 | 95.58 | 0.19 | 1.20 | 0.01 | 1.47 |
| | 1050 | 1.10 | 0.12 | 60.20 | 36.83 | 97.03 | 0.09 | 1.66 | 0.01 | 1.38 |
| 7 | 950 | 2.45 | 0.61 | 22.07 | 67.56 | 89.63 | 0.36 | 6.95 | 0.03 | 1.57 |
| | 1050 | 1.35 | 0.56 | 40.47 | 56.08 | 96.55 | 0.07 | 1.47 | 0.01 | 1.50 |
| 8 | 950 | 2.48 | 0.13 | 19.98 | 74.36 | 94.34 | 0.32 | 2.73 | 0.01 | 1.61 |
| | 1050 | 1.21 | 0.16 | 42.56 | 55.03 | 97.59 | 0.14 | 0.90 | 0.01 | 1.50 |
| 9 | 950 | 2.48 | 0.41 | 20.78 | 71.85 | 92.63 | 0.48 | 4.00 | 0.06 | 1.59 |
| | 1050 | 1.17 | 0.23 | 38.17 | 49.28 | 87.45 | 0.17 | 10.98 | 0.02 | 1.46 |
| 10 | 950 | 3.04 | 0.54 | 26.68 | 66.04 | 92.72 | 0.38 | 3.32 | 0.08 | 1.55 |
| | 1050 | 1.16 | 0.48 | 52.39 | 41.84 | 94.23 | 0.13 | 4.00 | 0.02 | 1.41 |
| 11 | 950 | 2.47 | 0.24 | 22.19 | 68.68 | 90.87 | 0.48 | 5.94 | 0.02 | 1.57 |
| | 1050 | 1.19 | 2.97 | 41.43 | 41.27 | 82.7 | 0.14 | 13.00 | 0.01 | 1.41 |
| 12 | 950 | 2.45 | 0.16 | 22.39 | 68.62 | 91.01 | 0.47 | 5.91 | 0.04 | 1.57 |
| | 1050 | 1.51 | 1.71 | 45.86 | 42.23 | 88.09 | 0.16 | 8.53 | 0.01 | 1.41 |
| 13 | 950 | 2.21 | 2.28 | 26.49 | 56.28 | 82.77 | 0.44 | 12.30 | 0.01 | 1.50 |
| | 1050 | 1.42 | 0.90 | 52.80 | 38.76 | 91.56 | 0.15 | 5.9 | 0.01 | 1.39 |
| 14 | 950 | 2.21 | 0.03 | 33.26 | 62.08 | 95.34 | 0.43 | 1.99 | 0.01 | 1.53 |
| | 1050 | 1.43 | 0.04 | 57.02 | 39.81 | 96.83 | 0.18 | 1.52 | 0.01 | 1.39 |
| 15 | 950 | 2.07 | 0.03 | 43.48 | 51.95 | 95.43 | 0.29 | 2.18 | 0.02 | 1.47 |
| | 1050 | 1.67 | 0.03 | 73.02 | 22.06 | 95.08 | 0.09 | 3.13 | 0.01 | 1.27 |
| 16 | 950 | 2.17 | 0.04 | 43.97 | 51.45 | 95.42 | 0.26 | 2.11 | 0.02 | 1.46 |
| | 1050 | 1.61 | 0.03 | 71.48 | 24.11 | 95.59 | 0.08 | 2.69 | 0.01 | 1.29 |
| 17 | 950 | 2.19 | 0.11 | 23.04 | 72.64 | 95.68 | 0.29 | 1.73 | 0.01 | 1.60 |
| | 1050 | 1.18 | 0.01 | 36.18 | 62.10 | 98.28 | 0.14 | 0.39 | 0.00 | 1.54 |
| 18 | 950 | 2.42 | 0.02 | 23.65 | 70.64 | 94.29 | 0.36 | 2.91 | 0.01 | 1.59 |
| | 1050 | 1.40 | 0.01 | 43.44 | 54.86 | 98.3 | 0.25 | 0.04 | 0.01 | 1.49 |

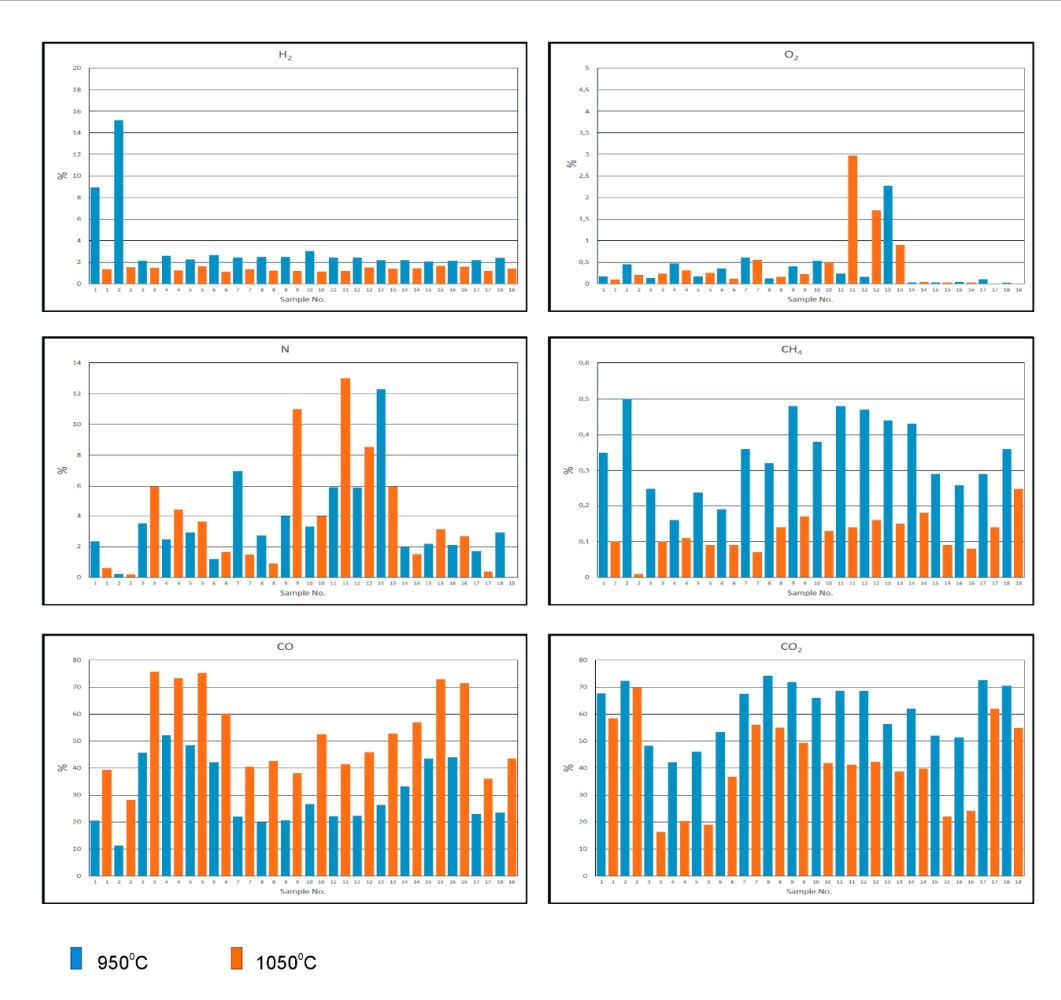

**Figure 6.** Syngas composition at temperatures up to 950 °C and 1050 °C.

The obtained results show clear differences in relation to the products obtained during gasification under an oxygen or air atmosphere [40,41]. Compared to the results from the literature [42,43], the experiment resulted in higher CO and $CO_2$ contents in the syngas. This is related to the supply of additional $CO_2$ to the system. As shown by numerous studies [44,45], introducing carbon dioxide into the gasification system allows us to lower the effective gasification temperature. However, the reaction rate with steam was higher than that of carbon dioxide gasification [46].

Various coal properties have an impact on the reactivity and quantities of released gas. It is commonly assumed that the reactivity of coal decreases with the increasing rank of coal, while it may vary for coals of lower ranks [47,48].

The gasification of different coals carried out by Liu et al. [49] in entrained flow gasifiers showed that the reactivity is not dependent on the degree of coalification measured by the fixed carbon or the volatile content, while the impact of the content and composition of the mineral matter is limited. In turn, tests carried out in fluidized bed gasifiers confirmed a correlation between the content of alkaline elements and reactivity [50].

The pyrolysis of coal is one of the gasification stages. Therefore, its key parameters, such as volatile content and porosity, are extremely important in determining the reactivity [51]. It was found that the reactivity of coal increases with the temperature [52]. Therefore, when it comes to gasification in the experimental reactor, where temperatures are relatively low (800–1100 °C), the use of highly reactive coal is required. It has been confirmed that the active site concentration in the molecular

structure of char is affected by both the aromaticity and degree of molecular ordering in the parent maceral structure [53].

On the other hand, it is worth noting that coal is a mixture of coalified organic matter, including macerals and mineral matter. The petrographic composition of coal is crucial for the gasification process, because different macerals have different reactivity [54–56]. The behavior of different macerals, namely vitrinite, liptinite, and micrinite, on heating at temperatures up to 800 °C has shown that the highest yield of volatiles can be obtained from liptinite, while the lowest can be obtained from micrinite [28].

The findings of the proximate and ultimate analysis were correlated. Petrographic determinations were also performed during the analysis. Table 4 presents Pearson correlations with significant relationships ($p$ = 0.05). Vitrinite (usually the main coal component) affects the majority of industrial processes. In the case of medium rank coals, it has relatively easy ignition characteristics; this also affects the course and products of the gasification process [57]. Vitrinite has a generally high oxygen content in contrast to other macerals. Carbon increases and oxygen decreases steadily during coalification, while the highest hydrogen content (about 85% C) can be found in vitrinite [58,59]. Vitrinite is rich in aromatic structures[3]. The aromaticity grows with the degree of coalification. The aromaticity increases with decreasing H/C and O/C atomic ratios, and therefore more energy is required to break these bonds. As a result, the gasification process is more difficult for coals of a higher rank. The composition and aromaticity of collotelinite also vary depending on the rank of the coal. An increase in aromaticity results in an increase in reflectance. Nevertheless, it is worth noting that the hydrogen content of collotellinite in coal with a reflectance between 0.5 and 1.6% is around 5% [60]. Collotelinite is the principal reactive maceral in technological processes, including carbonization and liquefaction. Yet, the reactivity of vitrinite in carbonization is limited to a random reflectance ranging from around 0.8–1.6% Rr, and less commonly up to 2.0%. The studies on the gasification and combustion reactivity of coals have confirmed that ignition temperatures and burnout rates depend on collotelinite reflectance [54]. The content of aliphatic substances in all of the macerals from the liptinite group is relatively high [61]. The ratio of aliphatic to aromatic components is the highest for the liptinite group. The chemical properties of resinite may vary significantly depending on precursor materials and early diagenetic modifications. Terpene resinite is generally composed of terpenoids. In addition, esters, phenols, alcohols, and resin acids can also be found [24]. Meanwhile, lipid resinite is mainly composed of fats and waxes. In terms of the chemical composition, the hydrogen content ranges from 8 to 11%. When it comes to other maceral groups, inertinite has a high carbon content and a low oxygen and hydrogen content [58]. In the case of semifusinite, its reflectance and structure are intermediate between vitrinite and fusinite, humotelinite/vitrinite, and fusinite [62]. This analysis has confirmed that different parameters affect the gas composition depending on temperature conditions (up to 950 and 1050 °C, respectively). At a temperature of 950 °C, the rank of coal (manifested by the vitrinite reflectance, carbon content, and volatile matter) has the highest impact on the $H_2$ content. The $H_2$ content in gas collected at a temperature of 950 °C increases with the increasing reflectance and carbon content and decreasing content of volatile matter. This relationship is not observed at 1050 °C. It can be concluded that the largest amounts of hydrogen are released at a temperature of about 950 °C, which is due to the chemical composition of coal [58]. Interestingly, it has been observed that the hydrogen content in gas decreases at both 950 °C and 1050 °C with the increasing hydrogen content in the heated coal. This negative correlation is due to the outlier value for the anthracite sample. During gasification at 950 °C, the hydrogen content in the gas was determined to be 15.18%, while the $H^{daf}$ content in the starting sample amounted to 3.17%. It was found that the hydrogen content in the gas at 950 °C increased dose dependently with the increase in the collotelinite content. This was related to the fact that hydrogen, in this component, was present in bonds of lower aromaticity and thus was released more easily at lower temperatures. On the other hand, the hydrogen content decreased with an increasing content of inertinite and microsporinite. The hydrogen content of inertinite was relatively low; thus, the greater the hydrogen content, the less $H_2$ in the syngas. At 950 °C, the $CO_2$ content

increased as CO and $CH_4$ proportions in the gas decreased with increasing $H_2$ content. These patterns were not observed at a temperature of 1050 °C.

**Table 4.** The relationship between proximate and ultimate parameters of all of the tested bituminous coal samples and the syngas composition.

| Temperature | 600–950 °C | | | | | | | 950–1050 °C | | | | | | |
|---|---|---|---|---|---|---|---|---|---|---|---|---|---|---|
| **Gas Component** | $H_2$ | $O_2$ | CO | $CO_2$ | $CH_4$ | $N_2$ | $H_2S$ | $H_2$ | $O_2$ | CO | $CO_2$ | $CH_4$ | $N_2$ | $H_2S$ |
| $M_t^{\,ar}$ [%] | - | - | 0.86 | −0.86 | −0.64 | - | - | - | - | 0.82 | −0.84 | - | - | - |
| $A^{db}$ [%] | - | - | 0.50 | - | −0.53 | - | - | - | - | - | - | - | - | - |
| $V^{daf}$ [%]f | −0.90 | - | 0.63 | −0.51 | - | - | - | - | - | 0.64 | −0.71 | - | - | - |
| $S_t^{\,db}$ [%] | - | - | - | - | - | - | - | - | - | - | - | - | - | 0.73 |
| NCV [MJ/kg] | - | - | - | - | - | - | - | - | - | - | - | - | - | - |
| $C^{daf}$ [%] | 0.66 | - | −0.83 | 0.75 | 0.50 | - | - | - | - | −0.85 | 0.90 | - | - | - |
| $H^{daf}$ [%] | −0.77 | - | - | - | - | - | - | - | - | - | - | 0.77 | - | - |
| $N^{daf}$ [%] | - | - | - | - | 0.60 | 0.58 | - | - | 0.63 | - | - | 0.55 | 0.71 | - |
| $O^{daf}$ [%] | −0.56 | - | 0.88 | −0.81 | −0.59 | - | - | - | - | 0.87 | −0.88 | - | - | - |
| *Petrographic composition [%]* | | | | | | | | | | | | | | |
| Telinite | - | - | - | - | - | - | −0.50 | - | - | - | - | - | - | - |
| Collotelinite | 0.77 | - | - | - | - | - | - | - | - | - | - | −0.55 | - | - |
| Collodetrinite | - | - | - | - | - | - | - | - | - | - | - | - | 0.49 | - |
| Vitrodetrinite | - | - | - | - | −0.51 | - | - | - | - | - | - | - | - | - |
| Corpogelinite | - | - | - | - | - | - | - | - | - | - | - | - | - | - |
| Gelinite | - | - | - | - | - | - | - | - | - | - | - | - | - | - |
| Macrosporinite | - | - | - | - | - | - | - | - | - | - | - | - | - | - |
| Microsporinite | −0.55 | - | - | - | - | - | - | - | - | - | - | 0.52 | - | - |
| Cutinite | - | - | - | - | - | - | - | - | - | - | - | - | - | - |
| Resinite | - | - | - | - | - | - | - | - | - | - | - | 0.63 | - | - |
| Liptodetrinite | - | - | - | - | - | - | - | - | - | - | - | - | - | - |
| Fusinite | - | - | - | - | - | - | - | - | - | - | - | - | - | - |
| Semifusinite | - | - | 0.61 | −0.51 | - | - | −0.47 | - | - | 0.55 | −0.48 | - | - | - |
| Funginite | - | - | - | - | - | - | - | - | - | - | - | - | - | - |
| Macrinite | - | - | - | - | - | - | - | - | - | - | - | - | - | - |
| Micrinite | - | - | - | - | - | - | - | - | - | - | - | - | - | - |
| Secretinite | - | - | - | - | - | - | - | - | - | - | - | - | - | - |
| Inertodetrinite | - | - | - | - | - | - | - | - | - | - | - | - | - | - |
| Sulfides | - | - | - | - | - | - | 0.66 | −0.47 | - | - | - | - | - | 0.66 |
| Carbonates | - | - | - | - | - | - | - | - | - | - | - | - | - | - |
| Clay minerals | - | - | - | - | - | - | - | - | - | - | - | - | - | - |
| Vitrinite | 0.52 | - | - | - | - | - | - | - | - | - | - | - | - | - |
| Liptinite | - | - | - | - | −0.51 | - | - | - | - | - | - | - | −0.50 | - |
| Inertinite | −0.47 | - | - | - | - | - | - | - | - | - | - | - | - | - |
| Mineral matter | - | - | - | - | - | - | - | - | - | - | - | - | - | - |
| Reflectance % | 0.96 | - | −0.55 | - | - | - | - | - | - | −0.55 | 0.61 | −0.49 | - | - |
| **Gas composition up to 950 °C [%]** | | | | | | | | | | | | | | |
| $H_2$ | - | - | - | - | - | - | - | - | - | −0.47 | 0.53 | −0.53 | - | - |
| $O_2$ | - | - | - | - | - | 0.79 | - | - | - | - | - | - | - | - |
| CO | - | - | - | −0.95 | −0.80 | - | - | - | - | 0.97 | −0.92 | - | - | - |
| $CO_2$ | - | - | −0.95 | - | 0.70 | - | - | - | - | −0.94 | 0.91 | - | - | - |
| $CH_4$ | - | - | −0.80 | 0.70 | - | - | - | - | 0.48 | −0.70 | 0.56 | - | - | - |
| $N_2$ | - | 0.79 | - | - | - | - | - | - | 0.53 | - | - | - | 0.48 | - |
| $H_2S$ | - | - | - | - | - | - | - | - | - | - | - | - | - | 0.87 |
| **Gas composition up to 1050 °C [%]** | | | | | | | | | | | | | | |
| $H_2$ | - | - | −0.47 | 0.53 | −0.53 | - | - | - | - | - | - | - | - | - |
| $O_2$ | - | - | - | - | 0.48 | 0.53 | - | - | - | - | - | - | 0.75 | - |
| CO | −0.47 | - | 0.97 | −0.94 | −0.70 | - | - | - | - | - | −0.96 | - | - | - |
| $CO_2$ | 0.53 | - | −0.92 | 0.91 | 0.56 | - | - | - | - | −0.96 | - | - | - | - |
| $CH_4$ | −0.53 | - | - | - | - | - | - | - | - | - | - | - | - | - |
| $N_2$ | - | - | - | - | - | 0.48 | - | - | 0.75 | - | - | - | - | - |
| $H_2S$ | - | - | - | - | - | - | 0.87 | - | - | - | - | - | - | - |

At a temperature of up to 950 °C, the CO content in the gas was mainly affected by the coal moisture, ash content, and volatile matter. In addition, lower amounts of CO in the gas produced from coals of a higher rank, which clearly shows a negative correlation between the CO content in the gas and the $C^{daf}$ and vitrinite reflectance, have also been recorded. It has been found that the semifusinite

content in coal increases with an increasing content of CO. Similar patterns were also observed for relationships between coal parameters and the CO content in gas collected at 1050 °C. The inverse relationship was observed for the $CO_2$ content. It has been found that the $CO_2$ content in the gas at temperatures of 950 and 1050 °C decreased with increasing CO content. The $CO_2$ content increased with an increasing rank of coal. This was particularly evident at 1050 °C, when the $C^{daf}$ increased with the $CO_2$ content.

With regards to the reduction in the CO and $CO_2$ generated, it should be stated that the amount of carbon dioxide obtained was inversely proportional to the content of $CO_2$ in the syngas. The dominant component of syngas is carbon dioxide or $CO_2$, which is obvious due to the use of $CO_2$ as a gasifying agent. However, some regularities were observed when it came to the carbon oxide content in syngas. What should be noted is the fact that smaller amounts of CO and $CO_2$ were produced during the gasification of coal from the Bogdanka deposit.

Coal from the Lublin Coal Basin has different properties in terms of gas generation compared to coal from the Upper Silesian Coal Basin. The high volatility of the gas-bearing capacity observed in the profile, and the volatile matter content increase with increasing depth, are especially worth noting. This was probably due to the diversity of the petrographic composition of coal in seams, which was dependent on sedimentation conditions. This is particularly evident in differently developed limo-fluvial and paralic sediments [63]. The Carboniferous deposits of the Lublin Coal Basin are known for methane–nitrogen gas-bearing coal, while the amount of $CO_2$ in the composition of these gases is low [64].

The methane content in the gas is clearly associated with the temperature. It has been found that at a temperature of up to 950 °C the decrease in the $CH_4$ content in the gas is dependent on the moisture content and ash content. The $CH_4$ content in coal decreased with an increasing content of vitrodetrinite and liptinite. On the other hand, a positive relationship between the microsporinite and resinite content and the $CH_4$ content in the collected gas samples was observed at a temperature of 1050 °C. This was most likely caused by the production of $CH_4$ from macerals from the liptinite group, which took place at a higher temperature than when the experiment was conducted in a $CO_2$ atmosphere.

The $H_2S$ content in gas is mainly associated with sulfur content and sulfides in the examined coal.

The gasification rate and carbon conversion efficiency increased with increasing temperature of the process. Furthermore, this effect was even more highlighted in heterogeneous $H_2O/CO_2$ atmospheres, than for respective homogeneous atmospheres. The gasification reactivity of chars from lower rank coals, which were less sensitive to the gasification temperature, was better. Despite the fact that the duration of the gasification process is primarily related to the rank of coal, it may be reduced by more than 50% as a result of an increase in the gasification temperature [33].

The characteristics of the process change depending on the atmosphere of gasification. When it comes to factors related to the gasification efficiency, the lowest values are recorded in a pure $CO_2$ atmosphere [65–69].

## 4. Conclusions

The most important phenomenon recorded during the analysis was pyrolysis. Gasification begins at a temperature of 900 °C. Pyrolysis is related to the distillation of organic substances under oxygen-free conditions. The examined coals were gasified under a $CO_2$ atmosphere. The temperature is clearly the most important variable that affects pyrolysis. Therefore, it has a direct impact on the resulting char, liquids, and gases. As a result of the conducted experiment involving the gasification of coal at temperatures of up to 1100 °C, we found that the amount of released gas mainly depends on the rank of coal. Furthermore, two phases, related to pyrolysis and gasification, were determined. The pyrolysis phase is dominant in the case of coking coal and anthracite; when it comes to coals of lower rank, gases are generally released during the gasification process. During the study, no clear correlation between the lithotype of coal and gas density was confirmed. The impact of individual macerals on the composition of gases at different temperatures was observed. We showed that they affect mainly

the chemical composition and the amount of aromatic and aliphatic components in the individual macerals. We also confirmed that, at a temperature of up to 950 °C, the increased content of collotelinite increases the $H_2$ content; on the other hand, at a temperature of 1050 °C it reduces the $CH_4$ content in the gas. In contrast, an increase in the microsporinite content in a temperature of up to 950 °C results in a decreased content of $H_2$, while in the case of a temperature of 1050 °C it contributes to a higher $CH_4$ content. The decrease in $CH_4$ content depends on increased amounts of liptinite. When it comes to the syngas composition, we found that larger quantities of macerals from the inertinite group, mostly semifusinite, are the most unfavorable. The mentioned macerals contribute to an increase in the content of unfavorable carbon dioxide and CO. Based on this analysis, we found that the selection of gasification technology should be adapted to the gasified fuel. Coal with a vitrinite reflectance of up to 0.6% releases the greatest amount of gases at temperatures of up to 1100 °C. The reaction of coal with the surrounding gas is a superficial process: the finer the coal, the greater the surface availability. The temperature increase is accompanied by phenomena affecting the surface area of the reaction: the degasification and formation of coke from ortho-bituminous coal contributes to its development, while the initial stages of coking coal types lead to sintering and the formation of agglomerates of grains. For this reason, no intensive gasification or gas release at higher temperatures for coals of higher rank is observed. Due to the prospects and possibilities of introducing the gasification process, the deposits in the Lublin Coal Basin (LCB) are the most promising. The coal from these deposits is characterized by a lower carbon oxides content in the obtained syngas, which makes the gasification process more environmentally friendly. There is only one mine in the Lublin Coal Basin; the Bogdanka coal mine. The prospective area of the deposit is approximately 9100 km$^2$, while documented deposits occupy an area of approximately 1200 km$^2$. The only currently active coal mine in LCB exploits one deposit, Bogdanka, while the LCB deposit in the K-3 area is in the development stage. Both deposits have a total area of approximately 92 km$^2$, which is 0.9% of the entire basin. There are industrial resources of 257,000,000 Mg, and an annual production of 6,924,000 Mg [70]. The location of the gasification plant in the Bogdanka area is therefore justified due to the large coal reserves occurring in the LCB, and the easier exploitation of the seams compared to the Upper Silesian Coal Basin.

**Author Contributions:** Conceptualization, B.B. and J.M.; Methodology, B.B. and J.M.; Formal Analysis, B.B. and J.M.; Data Curation, B.B.; Writing—Original Draft Preparation, B.B.; Writing—Review & Editing, B.B and J.M.; Visualization, J.M. All authors have read and agreed to the published version of the manuscript.

**Funding:** This research was funded by the Polish National Science Centre under research project awarded by decision no. DEC-2013/09/D/ST10/04045 and from subsidy no 16.16.140.315.

**Acknowledgments:** The authors would like to thank Wojciech Szulik and Karol Szymura from CLP-B in Jastrzębie-Zdrój for their help during the study.

**Conflicts of Interest:** The authors declare no conflict of interest.

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
