# Peer review of "The Impact of Coal’s Petrographic Composition on Its Suitability for the Gasification Process: The Example of Polish Deposits"

_resources, doi:10.3390/resources9090111_

Round 1

Reviewer 1 Report

This is a good paper, however, the level of English should be improved.

Author Response

Thank you for your positive review of the manuscript. The English language was checked by English Editing Publishing Services.

Reviewer 2 Report

The authors investigated experimentally the petrographic composition and technological parameters of coal on the gasification process.

Here is a list of my specific comments:

  1. more detailed discussion on previous studies as well as a clear statement on the novelty of the present work should be included in the introduction section.
  2. References for data and statistics provided in the introduction section should be cited.
  3. The results section is poorly written and the presentation of data lacks clarity:
  1. Legends in figures 3-5 need clear explanations.
  2. Values of horizontal and vertical axes in figures 3-5 are not shown.
  3. In table 3, the two cells for each sample number should be merged to show each sample number is represented by two rows.
  4. Axis titles in figure 6 should be inserted.
  5. The authors should either use the word “Figure” or “Fig.”
  6. the mass of detail and numbers presented outside of tables and graphs makes it difficult to focus on the important information. A rewrite of the paper to discard distracting detail and clarify key elements is necessary.

Author Response

The authors would like to thank for valuable comments and linguistic corrections. Some of the specific comments need to be answered:

The authors investigated experimentally the petrographic composition and technological parameters of coal on the gasification process.

Here is a list of my specific comments:

  1. more detailed discussion on previous studies as well as a clear statement on the novelty of the present work should be included in the introduction section.

Several references on CO2 gasification have been added.

It also has been stated that the novelty of the presented work is that the authors compared the suitability of coals with different petrographic compositions for gasification under CO2 atmosphere.

  1. References for data and statistics provided in the introduction section should be cited.

References have been added to the manuscript.

  1. The results section is poorly written and the presentation of data lacks clarity:
  1. Legends in figures 3-5 need clear explanations.

Values of horizontal and vertical axes in figures 3-5 are not shown.

Thank you for highlighting the problem with figures. This is due to Import/Export errors when working with Excel. In order to avoid problems with the import of figures, they were saved in a graphic format. Descriptions of sample numbers have also been added.

  1. In table 3, the two cells for each sample number should be merged to show each sample number is represented by two rows.
  2. Axis titles in figure 6 should be inserted.
  3. The authors should either use the word “Figure” or “Fig.”

Done.

  1. the mass of detail and numbers presented outside of tables and graphs makes it difficult to focus on the important information. A rewrite of the paper to discard distracting detail and clarify key elements is necessary.

Of course, we agree that there are many numbers in the text; however, the manuscript discusses the possibility of using statistical tools to assess the suitability of coal for gasification and unfortunately it was necessary to include individual parameters in the text. The discussion of the results would be incomplete without references to the tables in the text.

The manuscript has been rewritten. The English language was checked by English Editing Publishing Services.

 We hope that the revised manuscript is more understandable.

Reviewer 3 Report

This paper proposes a novel low-temperature coal gasification technology, which uses carbon dioxide as a gasification agent to gasify coal, and explores the effects of coal ranks, petrographic composition, and coal physical and chemical properties on the release and composition of synthesis gas. This research has relatively great practical value:

  1. Since carbon dioxide is proposed as a gasification agent, it is necessary to compare with ordinary gasification agents to show how much the gasification products have changed.
  2. How much carbon dioxide does the coal gasification process need to consume and how economical is it?
  3. The numbers 1, 2, 3, 4... in Figure 3-5 indicate what needs to be marked in the figure.
  4. The coordinate axis of Figure 3-5 needs to be scaled, and the two stages should also be marked in the figure.
  5. The format of the form is wrong, and a three-line format is required.
  6. The table cannot visually see the difference between different coal gasification products. It is necessary to add a figure to compare the same main gas products of several coals.
  7. What coal is used in Table 4 should be written in the header.
  8. The conclusion is too long.

Author Response

The authors would like to thank for valuable comments and linguistic corrections. Some of the specific comments need to be answered:

This paper proposes a novel low-temperature coal gasification technology, which uses carbon dioxide as a gasification agent to gasify coal, and explores the effects of coal ranks, petrographic composition, and coal physical and chemical properties on the release and composition of synthesis gas. This research has relatively great practical value:

  1. Since carbon dioxide is proposed as a gasification agent, it is necessary to compare with ordinary gasification agents to show how much the gasification products have changed.

We did not simultaneously conduct gasification using other gasifying agents, e.g. oxygen or air; therefore, we are not able to state how the gasification products have changed. The comparison with the results obtained by other authors is also difficult because gasification products depend on many factors, which was discussed in the introduction.

However, in the discussion of the results section, we referred to the differences between the products obtained during gasification under CO2, oxygen, and air atmosphere.

  1. How much carbon dioxide does the coal gasification process need to consume and how economical is it?

The inlet gas flow was set at 15 dm3/ h. - this information has been included in the description of the experiment. 

At this point, we are not able to say whether the entire process is economically feasible because the research was conducted on small samples as an experimental research. However, as shown by the results of the project on coal gasification under CO2 atmosphere that is conducted in Poland, the process is economically viable. 

  1. The numbers 1, 2, 3, 4... in Figure 3-5 indicate what needs to be marked in the figure.

Descriptions of the numbers in the figures have been added.

  1. The coordinate axis of Figure 3-5 needs to be scaled, and the two stages should also be marked in the figure.

The scales have been added. The stages were not marked so as to make the figures clear enough to read. We believe that the two stages are clearly visible.

  1. The format of the form is wrong, and a three-line format is required.

The tables have been corrected

  1. The table cannot visually see the difference between different coal gasification products. It is necessary to add a figure to compare the same main gas products of several coals.

The percentage share of individual syngas components for all samples is given in Fig. 6

  1. What coal is used in Table 4 should be written in the header.

All 18 bituminous coal samples were subjected to the correlation analysis. This information has been included in the header.

  1. The conclusion is too long.

We believe that we have included only the most important conclusions in the Conclusions. Because of the large number of results, the conclusions are also quite extensive, but they are only ¾ pages long.

The manuscript has been re-edited. The English language was checked by English Editing Publishing Services.

Round 2

Reviewer 3 Report

accept